# Extensive Synovial Chondromatosis of the Temporomandibular Joint Extending to the Cranial Base

**DOI:** 10.3390/diagnostics14202311

**Published:** 2024-10-17

**Authors:** Chi-Heon Sung, Seo-Young An, Hae-Seo Park, Wonae Lee, Moon-Young Kim

**Affiliations:** 1Department of Oral and Maxillofacial Surgery, College of Dentistry, Dankook University, Cheonan 31116, Republic of Korea; kom4809@gmail.com (C.-H.S.); u8042222@naver.com (S.-Y.A.); haeseopk@dankook.ac.kr (H.-S.P.); 2Department of Pathology, College of Medicine, Dankook University, Cheonan 31116, Republic of Korea; walee@dankook.ac.kr

**Keywords:** head and neck neoplasms, synovial chondromatosis, middle cranial fossa, TMJ

## Abstract

A 42-year-old male presented to the Department of Oral and Maxillofacial Surgery with the chief complaint of pain and stiffness in the right temporomandibular joint (TMJ). The patient’s height was 174 cm and his body weight was 65 kg. The patient’s occupation was heavy equipment operator. According to the patient, the pain had initiated a week prior to his first visit and was exacerbated during mastication. Evaluation of the range of motion revealed extensive crepitus along the right TMJ. The active and passive range of motion were measured at 45 mm and 42 mm, respectively, indicating adequate mouth-opening capacity. Occlusion was also favorable, and no other clinical symptoms were shown intraorally.

Synovial chondromatosis in the TMJ may occur as either primary idiopathic lesions or secondary lesions related to trauma or degenerative arthritis [1]. Patients often report discomfort within the TMJ that is exacerbated by physical activities such as mouth opening and mastication, thus resulting in TMJ dysfunction. Since the risk of degenerative arthritis resulting from constant mechanical friction increases with delay in the surgical removal of synovial chondromatosis, it is necessary to perform surgical excision as soon as the initial diagnosis is made.

On magnetic resonance imaging (MRI), synovial chondromatosis of the temporomandibular joint (TMJ) is indicated by the presence of cartilaginous nodules with signs of low and isointensity in the joint space, which may present small or localized rounded shapes, with different degrees of calcification, similar to a “ring” [2]. On computed tomography (CT), multiple loose bodies of moderate to increased density can be identified, delimited, and located, with expansion of the intra-articular space and bone changes, such as erosions of the mandibular fossa, joint tubercle, and head of the mandible [2].

Synovial chondromatosis is an uncommon articular disorder characterized by synovial metaplasia with intra-articular proliferation of cartilaginous nodules originating from the synovial membrane [3,4,5]. This disorder usually affects large joints and is rarely observed in the TMJ. Synovial chondromatosis in the TMJ may occur as either primary idiopathic lesions or secondary lesions related to trauma or degenerative arthritis [6].

Panoramic radiography, CT, and MRI are the typical radiological evaluation tools required for the diagnosis of TMJ bony lesions. However, a histopathological biopsy of the specimen acquired via surgery is crucial for definitive diagnosis. Synovial chondromatosis arising in the TMJ is most commonly reported to be confined within the joint cavity [7]. We herein report a rare extra-articular case of synovial chondromatosis extending to the cranial base. Panoramic radiographs taken upon the patient’s first visit revealed multiple calcified particles in the right TMJ area. CT and MRI were consecutively performed for accurate diagnosis. Radiological examinations revealed multiple nodules within the joint capsule of the right TMJ, extending to the middle cranial fossa. The shape of the mandibular condyle appeared to be smooth. The patient was provisionally diagnosed with pigmented villonodular synovitis and subsequently admitted for surgical treatment (Figure 1, Figure 2 and Figure 3).

Blood tests showed normal results with a white blood cell count of 9560/mm^3^, C-reactive protein at 0.36 mg/dL, and an erythrocyte sedimentation rate of 3 mm/h. Under general anesthesia, a temporo-preauricular incision was made to access the right temporomandibular joint, and surgical excision of the loose bodies was performed. A total of 25 loose bodies were removed, and histopathological examination was performed. At the 1-month post-surgery follow-up, the patient showed decreased pain, with a measured range of motion (ROM) of 35 mm. At the 6-month post-surgery follow-up, the patient still complained of occasional clicking in the right TMJ but stated that the pain had completely resolved. A follow-up CT scan taken at 6 months post-surgery also revealed no signs of loose body formation.

Synovial chondromatosis is a rare, benign condition of unknown etiology in which the synovium undergoes metaplasia leading to cartilaginous nodules that ultimately break free, mineralize, and even ossify [8]. This disorder usually affects large joints and is rarely observed in the TMJ [1]. The process of hyaline cartilage degeneration within the TMJ was originally described by Paré in 1558. Although the exact etiology remains yet unclear, it is characterized pathologically by degeneration of the synovial membrane within the articular space that results in the formation of calcified nodules [3]. The mean age of presentation is between the 4th and 5th decade, with a preponderance of female cases and unilateral presentation [8].

The common symptoms of synovial chondromatosis of the TMJ are pain, restricted mandibular range of motion, crepitation, and deviation to the affected site [5]. Typically, synovial chondromatosis occurring in the TMJ is predominantly localized within the joint space. Among the rare cases in which the extra-articular calcified nodules remarkably erode the skull base, neurological manifestations only still occur when the cranial base is perforated. The objective of this paper is to review the risk factors that may cause synovial chondromatosis arising from the TMJ, which previously led to neurological symptoms, by sharing our rare case.

Lieger et al. reviewed 80 cases of TMJ synovial chondromatosis patients to find only 7 cases with lesions extending to the cranial base [9]. They reported no significant differences in the neurological symptoms between this group and other cases with lesions not affecting the cranial base. However, neurological deficits (hearing loss) were identified when the loose body penetrated the temporal bone. It would be natural to assume that the delayed treatment of this condition would be one of the major causes leading to such an extension to the cranial base. However, von Lindern et al. reported contrary findings [10]. In their recent literature review of 60 references with a total of 74 cases, it was revealed that the mean delay to confirmation of diagnosis among patients with intracranial/cranial extension (21 months) was significantly shorter than that of extracranial cases (32 months). Age, gender, and the location of the lesion were similar in both groups. This case demonstrates the possibility of synovial chondromatosis of the TMJ extending to the base of the skull, underscoring the importance of the meticulous examination of the skull base in future diagnosis and treatment planning. In conclusion, we propose that a thorough evaluation of adjacent structures for signs of invasion be essentially accompanied when diagnosing TMJ synovial chondromatosis. Further research is necessary to elucidate the pathological mechanisms and clinical significance of such an invasion.

## Figures and Tables

**Figure 1 diagnostics-14-02311-f001:**
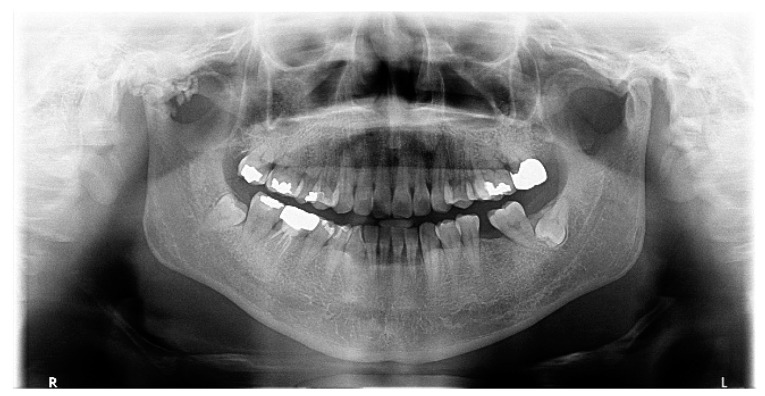
Panoramic view: multiple calcified nodules around the right condyle of the mandible, showing cortical disruption and possible erosion of the condyle head. The right condylar process shows a sclerotic appearance with mixed radiopacity and radiolucency. The joint space is reduced because of the lesion.

**Figure 2 diagnostics-14-02311-f002:**
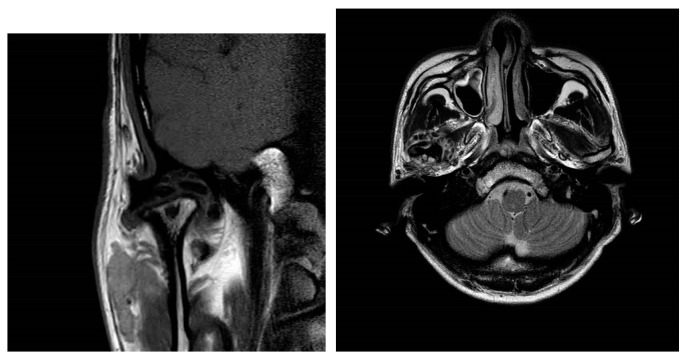
Coronal view of magnetic resonance (MR) TMJ T1-weighted image, showing a mass extending to the cranial base. The coronal view shows flattening of the right condylar head with a heterogeneous signal intensity, particularly in the cortical and subcortical areas, which may indicate sclerosis or degeneration. The articular disk of the right TMJ appears to be displaced anteriorly.

**Figure 3 diagnostics-14-02311-f003:**
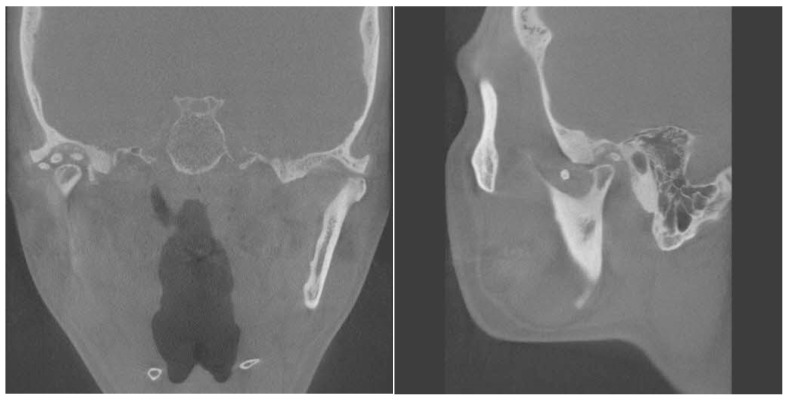
Coronal and sagittal view of computed tomography. Right glenoid fossa erosion was present compared to the opposite condyle. The right condylar head shows a loss of the normal smooth cortical margin with evidence of mixed sclerosis and radiolucency which shows cortical irregularity, and possible erosion is noted on the superior aspect of the condyle. The joint space is narrowed, suggesting possible degenerative changes.

## Data Availability

The original contributions presented in the study are included in the article, further inquiries can be directed to the corresponding author.

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
