# Peer review of "Extensive Synovial Chondromatosis of the Temporomandibular Joint Extending to the Cranial Base"

_diagnostics, 2024, doi:10.3390/diagnostics14202311_

Round 1

Reviewer 1 Report

Comments and Suggestions for Authors

I have a few remarks regarding the paper. First of all, citations for works no. 1-2 are missing.
There should be an introduction with references to images at the beginning, and a citation is necessary.
Please add an introduction that includes the following information:

• Provide a more detailed description of Synovial chondromatosis - 10.1016/j.omsc.2020.100144
• Highlight that Synovial chondromatosis rarely affects the TMJ - 10.1259/bjr/69067316
• Also add that the most common condition related to the TMJ is the Prevalence of Temporomandibular Disorders - 10.3390/jcm13051365

Additionally, when using abbreviations, you must define them the first time – this applies to the entire text.
The study lacks a clear objective.
The specifics of the patient are missing, such as weight and occupation – please provide more information about them.
Unfortunately, 7 out of 8 references are over 10-20 years old, which is unacceptable. Please supplement with more recent information.
Correct the references according to the journal's guidelines.

Author Response

Reviewer 1.

Comments 1: I have a few remarks regarding the paper. First of all, citations for works no. 1-2 are missing.

Response 1: We appreciate the reviewer for valuable advice and detailed suggestions. The numbering of citations was revised.

Comments 2: There should be an introduction with references to images at the beginning, and a citation is necessary.

Please add an introduction that includes the following information:

  • Provide a more detailed description of Synovial chondromatosis - 10.1016/j.omsc.2020.100144
  • Highlight that Synovial chondromatosis rarely affects the TMJ - 10.1259/bjr/69067316
  • Also add that the most common condition related to the TMJ is the Prevalence of Temporomandibular Disorders - 10.3390/jcm13051365

Response 2: We appreciate the reviewer for more information about our paper’s subject. We added more detailed information about synovial chondromatosis of the TMJ with the papers you gave us.

Comments 3: Additionally, when using abbreviations, you must define them the first time – this applies to the entire text.

Response 3: Thank you for your kind comment. Abbreviations were defined when used first time in this paper.

Comments 4: The study lacks a clear objective.

Response 4: We appreciate your comment. We added a clear objective in the conclusion.

Comments 5: The specifics of the patient are missing, such as weight and occupation – please provide more information about them.

Response 5: Thank you for your comment. The specifics about the patient were added (height, body weight, occupation).

Comments 6: Unfortunately, 7 out of 8 references are over 10-20 years old, which is unacceptable. Please supplement with more recent information.

Response 6: Thank you for your comment. We changed the Papers used for citation to recent ones, without compromising the content.

Reviewer 2 Report

Comments and Suggestions for Authors

Interesting article. About introduction: I recommend developing a more detailed analysis (supported by the literature) on the synovial condromatosis of the temporomandibular joint, such as intra-oral and extra-oral clinical signs and radiographic examinations necessary for correct diagnosis.  

Author Response

Comments 1: The paper is interesting and covers a rare case of synovial chondromatosis with cranial base involvement, which adds value to the literature on temporomandibular joint pathology. It would be beneficial to expand the introduction by including the latest studies on this condition to better situate the case in the context of current medical knowledge.

Response 1: We appreciate for your detailed revision and analysis about our paper.

While revision, we added more recent studies about ‘synovial chondromatosis’ and more details about the diagnosis of the patient’s disease and description about the images. Again, We truly appreciate your comment.

Comments 2: The methods are well-described, but more details on the diagnostic process that led to the decision for surgery could further strengthen the section. The radiological images are helpful, though their descriptions could be more detailed to aid interpretation for less experienced readers. The discussion could be enhanced by including details about the mechanisms leading to cranial base involvement and the implications of delayed treatment.

Response 2: We sincerely thank you for your comment. We added the content that [risk of degenerative arthritis resulting from constant mechanical friction increases with delay in the surgical removal of synovial chondromatosis, it is necessary to perform surgical excision as soon as the initial diagnosis is made] to mention that surgery was essential.

Comments 3: The conclusions are well-formulated, but they could be expanded with suggestions for future research or clinical practice to provide more concrete takeaways for the readers.

Response 3: Thank you for your comment. We added the following sentence for readers to participate in the study of these rare cases. “Further research is necessary to elucidate the pathological mechanisms and clinical significance of such an invasion.”

Reviewer 3 Report

Comments and Suggestions for Authors

The paper is interesting and covers a rare case of synovial chondromatosis with cranial base involvement, which adds value to the literature on temporomandibular joint pathology. It would be beneficial to expand the introduction by including the latest studies on this condition to better situate the case in the context of current medical knowledge. The methods are well-described, but more details on the diagnostic process that led to the decision for surgery could further strengthen the section. The radiological images are helpful, though their descriptions could be more detailed to aid interpretation for less experienced readers. The discussion could be enhanced by including details about the mechanisms leading to cranial base involvement and the implications of delayed treatment. The conclusions are well-formulated, but they could be expanded with suggestions for future research or clinical practice to provide more concrete takeaways for the readers.

Author Response

Comments 1: Interesting article. About introduction: I recommend developing a more detailed analysis (supported by the literature) on the synovial condromatosis of the temporomandibular joint, such as intra-oral and extra-oral clinical signs and radiographic examinations necessary for correct diagnosis.  

Response 1: We sincerely appreciate your revision. We added more detailed analysis and comments about synovial chondromatosis with more recent papers.

Round 2

Reviewer 1 Report

Comments and Suggestions for Authors

Thank you for your response. At the beginning, please include the text in the article, followed by the images. Take a look at other works: https://www.mdpi.com/journal/diagnostics/topical_collections/interesting_images

Not all of my comments have been addressed. Most importantly, the key one, which is the updating of references, has not been done. The work is still based on outdated articles.

Author Response

Comments 1: At the beginning, please include the text in the article, followed by the images. Take a look at other works: https://www.mdpi.com/journal/diagnostics/topical_collections/interesting_images

Response 1: We sincerely appreciate you with your revision. We changed the procedure of our paper; article at the beginning followed by the images.

Comments 2: Not all of my comments have been addressed. Most importantly, the key one, which is the updating of references, has not been done. The work is still based on outdated articles.

Response 2: Thank you for your comment. We changed the references to newly updated articles.

Round 3

Reviewer 1 Report

Comments and Suggestions for Authors

Thank you for your response. A small note please correct the references according to the requirements of the journal.